# Deep geometric matrix completion: Are we doing it right?

## Abstract

We address the problem of reconstructing a matrix from a subset of its entries. Current methods, branded as *geometric matrix completion*, augment classical rank regularization techniques by incorporating geometric information into the solution. This information is usually provided as graphs encoding relations between rows/columns. In this work we propose a simple spectral approach for solving the matrix completion problem, via the framework of functional maps. We introduce the *zoomout loss*, a multiresolution spectral geometric loss inspired by recent advances in shape correspondence, whose minimization leads to state-of-the-art results on various recommender systems datasets. Surprisingly, for some datasets we were able to achieve comparable results even without incorporating geometric information. This puts into question both the quality of such information and current methods' ability to use it in a meaningful and efficient way.

## 1 Introduction

Matrix completion deals with the recovery of missing values of a matrix from a subset of its entries,

$$\text{Find } \boldsymbol{X} \text{ s.t. } \boldsymbol{X} \odot \boldsymbol{S} = \boldsymbol{M} \odot \boldsymbol{S}. \tag{1}$$

Here $\boldsymbol{X}$ stands for the unknown matrix, $\boldsymbol{M} \in \mathbb{R}^{m \times n}$ for the ground truth matrix, $\boldsymbol{S}$ is a binary mask representing the input support, and $\odot$ denotes the Hadamard product. Since problem (1) is ill-posed, it is common to assume that $\boldsymbol{M}$ belongs to some low dimensional subspace. Under this assumption, the matrix completion problem can be cast via the least-squares variant,

$$\min_{\boldsymbol{X}} \text{ rank}\,(\boldsymbol{X}) + \frac{\mu}{2} \left\| (\boldsymbol{X} - \boldsymbol{M}) \odot \boldsymbol{S} \right\|_F^2. \tag{2}$$

Relaxing the intractable rank penalty to its convex envelope, namely the nuclear norm, leads to a convex problem whose solution coincides with (2) under some technical conditions (Candès & Recht, 2009). Another way to enforce low rank is by explicitly parametrizing $\boldsymbol{X}$ in factorized form, $\boldsymbol{X} = \boldsymbol{Y}\boldsymbol{Z}^\top$. The rank is upper-bounded by the inner dimensions of $\boldsymbol{Y}, \boldsymbol{Z}^\top$. Recent studies by Arora et al. (2019); Gunasekar et al. (2017), suggest that overparametrizing $\boldsymbol{X}$ as a product of several matrices results in a low rank matrix due to an implicit regularization induced by the gradient descent trajectories. According to (Gunasekar et al., 2017; Arora et al., 2019), given enough entries and independent of the depth, overparametrized *deep matrix factorization* (DMF) models are equivalent[1] to nuclear norm minimization. However, in the data poor regime these methods differ, and an implicit regularization brought forward by the dynamics of the gradient descent algorithm provides a stronger rank regularization. The deeper the network - the stronger the regularization.

Optimization approaches as described above can be thought of as instances of *self-supervised learning*, an informal name given to a machine learning paradigm with the goal of "predicting any part of the input from any other part" (LeCun, 2019). Adopting the machine learning nomenclature, we shall henceforth refer to the given entries of $\boldsymbol{M}$ as *training set* or *training samples*, and denote by *test set* a subset of the (unknown) entries of $\boldsymbol{M}$ which we shall use for evaluation.

The advent of deep learning platforms equipped with efficient automatic differentiation tools allows the exploration of more sophisticated models that incorporate intricate regularizations, both explicit

---

[1] Under some technical conditions. See (Arora et al., 2019).

and implicit. Some contemporary approaches for matrix completion fall under the umbrella of *geometric deep learning*. These approaches generalize standard (Euclidean) deep learning to domains such as general graphs and manifolds. For example, *graph convolutional neural networks* (GCNNs) follow the architecture of standard CNNs, but replace the Euclidean convolution operator with linear filters constructed using the graph Laplacian. We distinguish between graph based approaches which make use of the bi-partite graph structure of the rating matrix (e.g., Berg et al. (2017)), and *geometric matrix completion* techniques which make use of side information in the form of graphs encoding relations between rows/columns (Kalofolias et al., 2014; Monti et al., 2017). While these techniques achieve state-of-the-art results, their design is arguably cumbersome and non-intuitive. It has recently been demonstrated that some graph CNN architectures can be greatly simplified, and still perform competitively on several graph analysis tasks (Wu et al., 2019). Such simple techniques have the advantage of being easier to analyze and reproduce. Motivated by these results, our intent is to construct a simple architecture for matrix completion that follows from geometric considerations rather than ad-hoc ones.

The inspiration for our paper stems from techniques for finding shape correspondence. In particular, the functional maps framework and its variants (Ovsjanikov et al., 2012; 2016). Most notably the work of (Litany et al., 2017) who combined functional maps with joint diagonalization to solve partial shape matching problems, and the *product manifold filter* (PMF) (Vestner et al., 2017b;a) and *zoomout* (Melzi et al., 2019) – two greedy algorithms for correspondence refinement by gradual introduction of high frequencies. This last method lent its name to the loss we define in Section 3. To that end, we propose a simple spectral method for geometric matrix completion that combines the fully linear network structure of DMF with the geometric framework of functional maps.

**Contribution.** Our contributions are as follows:

- We propose a geometric interpretation for *deep matrix factorization* (Arora et al., 2019) by embedding it as part of the functional maps framework.

- We introduce the *zoomout loss*, a multiresolution spectral geometric loss inspired by recent advances in shape correspondence.

- We show that via a simple *shallow and fully linear* network, it is possible to obtain state-of-the-art results on various recommendation systems datasets.

- We demonstrate that in some cases the effect of the geometry is only marginal, and results on par with state-of-the-art can be achieved even without it.

## 2 BACKGROUND

**Spectral graph theory.** Let $G = (V, E, \boldsymbol{\Omega})$ be a (weighted) graph specified by its vertex set $V$ and edge set $E$, and let $\boldsymbol{\Omega}$ be its adjacency matrix. Given a function $\boldsymbol{x} \in \mathbb{R}^{|V|}$ on the vertices, we define the following quadratic form (also known as *Dirichlet energy*) measuring the variability of the function $\boldsymbol{x}$ on the graph,

$$\boldsymbol{x}^\top \boldsymbol{L} \boldsymbol{x} = \sum_{(a,b) \in E} \omega_{a,b} \left( x(a) - x(b) \right)^2 . \tag{3}$$

The matrix $\boldsymbol{L}$ is called the *(combinatorial) graph Laplacian*, and is given by $\boldsymbol{L} = \boldsymbol{D} - \boldsymbol{\Omega}$, where $\boldsymbol{D} = \mathrm{diag}(\boldsymbol{\Omega 1})$ is the *degree matrix*. $\boldsymbol{L}$ is symmetric and positive semi-definite and therefore admits a spectral decomposition $\boldsymbol{L} = \boldsymbol{\Phi \Lambda \Phi}^\top$. The graph Laplacian is a discrete generalization of the continuous Laplace-Beltrami operator, and therefore has similar properties. One can think of the eigenpairs $(\boldsymbol{\phi}_i, \lambda_i)$ as the graph analogues of "harmonic" and "frequency". A function $\boldsymbol{x} = \sum_{i=1}^{|V|} \alpha_i \boldsymbol{\phi}_i$ on the vertices of the graph whose coefficients $\alpha_i$ are small for large $i$, demonstrates a "smooth" behaviour on the graph in the sense that the function values on nearby nodes will be similar. A standard approach to promoting such smooth functions on graphs is by using the Dirichlet energy (3) to regularize some loss term. For example, this approach gives rise to the popular bilateral and non-local means filters (Gadde et al., 2013).

**Functional maps.** Let $G_1 = (V_1, E_1, \boldsymbol{\Omega}_1)$, $G_2 = (V_2, E_2, \boldsymbol{\Omega}_2)$ be two graphs, and let $\boldsymbol{\Phi}, \boldsymbol{\Psi}$, be two orthonormal bases for functions defined on the vertices of these graphs. Given two such functions,

$x = \Phi\alpha$ on $G_1$ and $y = \Psi\beta$ on $G_2$, one can define a map $C$ between their representations $\alpha$ and $\beta$, i.e., $\alpha = \Phi^\top x = C\Psi^\top y = C\beta$. The matrix $C$ represents a linear map between the functional spaces on $G_1$ and $G_2$, known as a *functional map*. Let us denote the Cartesian product of $G_1$ and $G_2$ by $G_1 \square G_2$. $G_1 \square G_2$ is a graph with vertex set $V_1 \times V_2$, on which two nodes $(u, u'), (v, v')$ are adjacent if either $u = v$ and $(u', v') \in E_2$ or $u' = v'$ and $(u, v) \in E_1$. Let $X$ be a function defined on $G_1 \square G_2$, then the functional map is given by projecting $X$ onto the corresponding bases $\Phi, \Psi$, $C = \Phi^\top X \Psi$. Using the SVD, one can decompose $X = U\Sigma V^\top$ to interpret $C$ as a mapping between $U\Sigma$, a function on one graph, and $V$, a function on the other. To get $X$ back from the functional map, one can use $X = \Phi C \Psi^\top$. For computational reasons, it is common to use truncated bases $\Phi, \Psi$, in which case the last equality holds only approximately.

The structure of the functional map depends on the properties of the chosen bases and the functions it maps. A common choice for a basis is the aforementioned Laplacian eigenbasis, building on the assumption that the signals involved are smooth with respect to the graphs. While this is a useful model, it assumes that the given graphs encode the geometry in an adequate way. In real world problems these graphs are only approximate, constructed from heuristic features associated with the row and the column spaces. Given better graphs, a simpler structure of the functional map emerges. For example, by introducing two orthonormal matrices $P$ and $Q$, one can make the functional map $C = (\Phi P)^\top X (\Psi Q)$ diagonal. These orthonormal matrices can be thought of as a way of aligning the bases $\Phi, \Psi$, with the principal axes of $X$.

## 3 SPECTRAL GEOMETRIC MATRIX COMPLETION

We assume that we are given a set of samples from the unknown matrix $M \in \mathbb{R}^{m \times n}$, encoded by a binary mask $S$, and two graphs $G_r, G_c$, encoding relations between the rows and the columns, respectively. Denote the Laplacians of these graphs and their spectral decompositions by $L_r = \Phi \Lambda_r \Phi^\top$, $L_c = \Psi \Lambda_c \Psi^\top$. Our approach relies on a minimization problem of the form

$$\min_X E_z(X) + \mu_r \text{trace}\left(X^\top L_r X\right) + \mu_c \text{trace}\left(X L_c X^\top\right), \tag{4}$$

with $E_z$ denoting the data term that we discuss in the sequel. The other two terms in (4) are the Dirichlet energies of $X$ computed on $G_r, G_c$, as defined in (3). These energies measure the variability of $X$ along neighbouring nodes on these graphs. Using $X = \Phi C \Psi^\top$ and the spectral decompositions of $L_r, L_c$, we can equivalently write them in terms of $C$,

$$\begin{aligned}
\text{trace}\left(X^\top L_r X\right) &= \text{trace}\left(\Psi C^\top \Phi^\top L_r \Phi C \Psi^\top\right) = \text{trace}\left(C^\top \Lambda_r C\right), \\
\text{trace}\left(X L_c X^\top\right) &= \text{trace}\left(\Phi C \Psi^\top L_c \Psi C^\top \Phi^\top\right) = \text{trace}\left(C \Lambda_c C^\top\right).
\end{aligned} \tag{5}$$

As mentioned above, the input graphs are typically constructed from a set of heuristically gathered features which may provide a poor representation of the latent geometry. One way to account for this inaccuracy could be to include the features in our optimization. This will induce, through a complicated nonlinear dependence, a different metric (i.e., adjacency matrix) and a different graph Laplacian. This approach lies at the heart of smoothing models based on conditional random fields (CRFs) used in image segmentation (see for example Krähenbühl & Koltun (2011)). We adopt a different approach by working in the spectral domain. Switching to the spectral domain allows us to modify the metric indirectly by applying orthogonal transformations, $P$ and $Q$, to the bases $\Psi, \Phi$. The purpose of these transformations is to rotate $\Psi, \Phi$, in a way that will simplify the structure of $C$. For example, it is possible to diagonalize $C$ by an appropriate choice of $P, Q$ via the SVD decomposition.

Since our method relies on the premise that the matrix $M$ is smooth with respect to some graphs, our interest is in modified bases $\Phi P, \Psi Q$, which arise from the eigendecomposition of a graph Laplacian. To that end, we shall use $L_r, L_c$, as proxies for the latent graph Laplacians, and promote bases that approximately diagonalize them by introducing two energy terms,

$$\begin{aligned}
E_{\text{diag}}^r &\equiv \left\| \text{off}\left(P^\top \Phi^\top L_r \Phi P\right) \right\|_F^2 = \left\| \text{off}\left(P^\top \Lambda_r P\right) \right\|_F^2, \\
E_{\text{diag}}^c &\equiv \left\| \text{off}\left(Q^\top \Psi^\top L_c \Psi Q\right) \right\|_F^2 = \left\| \text{off}\left(Q^\top \Lambda_c Q\right) \right\|_F^2,
\end{aligned} \tag{6}$$

where $\mathrm{off}(\cdot)$ denotes the off-diagonal elements. Under these modifications, the Dirichlet energy terms (5) become,

$$
\begin{aligned}
E_{\mathrm{dir}}^r &\equiv \mathrm{trace}\left(\boldsymbol{Q}\boldsymbol{C}^\top \boldsymbol{P}^\top \boldsymbol{\Lambda}_{\mathrm{r}} \boldsymbol{P}\boldsymbol{C}\boldsymbol{Q}^\top\right), \\
E_{\mathrm{dir}}^c &\equiv \mathrm{trace}\left(\boldsymbol{P}\boldsymbol{C}\boldsymbol{Q}^\top \boldsymbol{\Lambda}_{\mathrm{c}} \boldsymbol{Q}\boldsymbol{C}^\top \boldsymbol{P}^\top\right).
\end{aligned}
\tag{7}
$$

Finally, we introduce the following energy terms to promote the (approximate) orthonormality of $\boldsymbol{P}, \boldsymbol{Q}$:

$$
\begin{aligned}
E_{\mathrm{orth}}^r(\boldsymbol{P}) &\equiv \left\|\boldsymbol{P}^\top \boldsymbol{P} - \boldsymbol{I}\right\|_F^2, \\
E_{\mathrm{orth}}^c(\boldsymbol{Q}) &\equiv \left\|\boldsymbol{Q}^\top \boldsymbol{Q} - \boldsymbol{I}\right\|_F^2.
\end{aligned}
\tag{8}
$$

**Zoomout loss.** Let us denote the training error achieved by a matrix $\boldsymbol{X}_{p,q} \equiv \boldsymbol{\Phi}_p \boldsymbol{C}_{p,q} \boldsymbol{\Psi}_q^\top$ composed from the first $p$ vectors in $\boldsymbol{\Phi}$ and the first $q$ vectors in $\boldsymbol{\Psi}$ as $Z_{p,q} \equiv \|\left(\boldsymbol{\Phi}_p \boldsymbol{C}_{p,q} \boldsymbol{\Psi}_q^\top - \boldsymbol{M}\right) \odot \boldsymbol{S}\|_F^2$. We define the zoomout loss as follows:

$$
E_{\mathrm{z}} = \sum_{p=1, q=1}^{m,n} w_{p,q} Z_{p,q},
\tag{9}
$$

with the weights $w_{p,q} \geq 0$. It can be shown that it is enough to use each value of $p$ or $q$ only once and therefore most of the $w_{p,q}$ shall be set to 0 (see Appendix A). The zoomout loss (9) is inspired by the greedy approaches for shape correspondence proposed by Vestner et al. (2017a) and Melzi et al. (2019). These methods approximate the point-to-point map between the shapes (i.e., the correspondence) by a smoothed map obtained through the application of spectral filters, and proceed by projecting the smoothed map onto the set of point-to-point maps. By gradually increasing the filters' bandwidth, or *resolution*, the correspondence becomes more refined in each step. In contrast, we advocate minimizing the training error using all resolutions at once. This simultaneous multi-resolution approach incurs a penalty on the rank of the reconstructed matrix by implicitly giving more weight to the low frequency terms. For example, the sub-matrix $\boldsymbol{C}_{2,2}$ appears in all the terms $Z_{p,q}$ with $p \geq 2, q \geq 2$, emphasizing its importance. This allows us to be sloppy in the estimation of the rank of $\boldsymbol{M}$ without compromising the results by much. We explore the effect of overparameterization in Section 4. Following the discussion above, we replace $\boldsymbol{\Phi}, \boldsymbol{\Psi}$ with $\boldsymbol{\Phi}\boldsymbol{P}, \boldsymbol{\Psi}\boldsymbol{Q}$, obtaining $Z_{p,q} \equiv \|\left(\boldsymbol{\Phi}\boldsymbol{P}_p \boldsymbol{C}_{p,q} \boldsymbol{Q}_q^\top \boldsymbol{\Psi}^\top - \boldsymbol{M}\right) \odot \boldsymbol{S}\|_F^2$. With this modification, the zoomout loss (9) should favor bases $\boldsymbol{\Phi}\boldsymbol{P}, \boldsymbol{\Psi}\boldsymbol{Q}$, in which most of the energy of $\boldsymbol{C}$ is concentrated in the low frequencies, i.e., the top-left part. An interesting observation is that by setting $p = m, q = n$, we get the *deep matrix factorization* (DMF) method from Arora et al. (2019) (up to initialization). As we show in Section 4, this model alone, without any additional geometric priors, is sufficient to obtain results on par with the state-of-the-art on some datasets. This puts in question the quality of the geometric information in those cases. The complete minimization objective combines all the described terms weighed with the appropriate weights and minimized with respect to $\boldsymbol{C}, \boldsymbol{P}$ and $\boldsymbol{Q}$ by gradient descent,

$$
\min_{\boldsymbol{C}, \boldsymbol{P}, \boldsymbol{Q}} E_{\mathrm{z}} + \mu_r E_{\mathrm{dir}}^r + \mu_c E_{\mathrm{dir}}^c + \rho_r E_{\mathrm{diag}}^r + \rho_c E_{\mathrm{diag}}^c + o_r E_{\mathrm{orth}}^r + o_c E_{\mathrm{orth}}^c.
\tag{10}
$$

### 3.1 THEORETICAL DISCUSSION

In this section we discuss some physical interpretations of our approach.

**Forward/Backward diffusion.** The backbone of the zoomout loss are the terms $Z_{p,q}$, which measure the disparity (restricted to the training set) between $\boldsymbol{M}$ and its smoothed version, $\boldsymbol{X}_{p,q} \equiv \boldsymbol{\Phi}_p \boldsymbol{C}_{p,q} \boldsymbol{\Psi}_q^\top$. While we rely on the full spectral decomposition of $\boldsymbol{L}_\mathrm{r}, \boldsymbol{L}_\mathrm{c}$ in order to construct $\boldsymbol{X}_{p,q}$ at different resolutions, one can also construct them gradually. The process of *zooming out*, i.e., switching from $\boldsymbol{X}_{p,q}$ to $\boldsymbol{X}_{p+1,q+1}$, consists of two steps: *smoothing* or *filtering* - applying spectral filters $\boldsymbol{F}_p, \boldsymbol{G}_q$, to the functional map $\boldsymbol{C} = \boldsymbol{\Phi}^\top \boldsymbol{X}\boldsymbol{\Psi}$,

$$
\boldsymbol{C}_{p,q} = \boldsymbol{F}_p \left(\boldsymbol{\Phi}^\top \boldsymbol{X}\boldsymbol{\Psi}\right) \boldsymbol{G}_q^\top,
\tag{11}
$$

and *sharpening* - forming $C_{p+1,q+1}$ from $C_{p,q}$ by adding the missing frequencies, and composing $X_{p+1,q+1}$ via

$$X_{p+1,q+1} = \Phi C_{p+1,q+1} \Psi^\top. \tag{12}$$

In the algorithm described above we chose $F_p, G_q$ to be diagonal matrices with $p$ or $q$ 1's on the diagonal, i.e., step filters. One can think about different kind of filters that, at least conceptually, will have a similar effect. For example, *smoothing* can be done with filters of the following form,

$$F_p = I - \tau_f L_{\mathrm{r}}, \quad G_q = I - \tau_f L_{\mathrm{c}}, \tag{13}$$

which correspond to a short time (forward) diffusion process, followed by a *sharpening* stage

$$(I + \tau_b L_{\mathrm{r}})(I - \tau_f L_{\mathrm{r}}) X (I - \tau_f L_{\mathrm{c}})(I + \tau_b L_{\mathrm{c}}), \tag{14}$$

which corresponds to backward diffusion. Forward-and-Backward (FAB) diffusion has been used for image denoising either as a nonlinear process on the original image grid (Gilboa et al., 2002), or via a linear process on a graph (Singer et al., 2009), and in shape processing for finding correspondence between non-rigid shapes (Vestner et al., 2017a). Despite backward diffusion being notoriously unstable, our algorithm exhibits a stable behaviour. In light of our experiments, we conjecture that it is possible to perform a stable backward diffusion on the product graph $G_{\mathrm{r}} \times G_{\mathrm{c}}$ via a controlled forward-and-backward diffusion process on each of the graphs $G_{\mathrm{r}}$ and $G_{\mathrm{c}}$. Notice that (14) is no more than a second degree polynomial of the Laplacian applied to the signal $X$ on both sides. Concurrent application of such polynomials result in a deep linear graph neural network. Our results and analysis support recent works such as Arora et al. (2019) and Wu et al. (2019) that argue in favor of such fully linear deep networks.

**Multirate filtering.**   Let us look at the gradient of $Z_{p,q} = \left\| \left( \Phi_p C \Psi_q^\top - M \right) \odot S \right\|_F^2$,

$$\frac{\partial Z_{p,q}}{\partial C} = \Phi_p^\top \left( \left( \Phi_p C \Psi_q^\top - M \right) \odot S \odot S \right) \Psi_q = \Phi_p^\top \left( R \odot S \right) \Psi_q. \tag{15}$$

In (15) we denoted by $R \equiv X_{p,q} - M$ the residual, and used the fact that $S \odot S = S$ for a binary mask. Ideally, the gradient should be the filtered version of the full residual $R$, but we only have its sampled version $R \odot S$. Is (15) a good approximation to $\Phi_p^\top R \Psi_q$?

From a signal processing perspective, it is sometimes possible to exchange the order of filtering and decimation (down-sampling). The conditions under which it is possible are known as the Noble identity for down sampling. Informally, if the down-sampler comes before the filter, the order between the filter and the down-sampler can be exchanged, but the filter has to be modified by interleaving it with zeros between adjacent samples. However, if the filter comes first, the order can only be exchanged if it has this special "interleaved with zeros" sparse form. Effectively, if the filter is a low-pass filter, it is possible to perform the down-sampling before the filtering. Since, by virtue of (5), our $X$ is driven towards having mostly low-frequencies, we can expect that the functional map obtained by the gradient descent iteration will be a good approximation of the true functional map[2].

## 4   RESULTS

We demonstrate the effectiveness of our approach on the following datasets: Synthetic Netflix, Flixster, Douban, Movielens (ML-100K) and Movielens-1M (ML-1M) as referenced in Table 1.

The datasets include user ratings for items (such as movies) and additional features. For all the datasets we use the users and items graphs taken from Monti et al. (2017). The ML-1M dataset was taken from Berg et al. (2017). We constructed 10 nearest neighbor graphs for users/items from the features, and used a Gaussian kernel with $\sigma = 1$ for edge weights. See Table 4 in the Appendix for a summary of the dataset statistics. For all the datasets, we report the results for the same test splits as that of (Monti et al., 2017) and (Berg et al., 2017). The compared methods are referenced in Table 1.

Synthetic Netflix is a small synthetic dataset constructed by Kalofolias et al. (2014) and Monti et al. (2017), in which the user and item graphs have strong communities structure. See Figure 6 in Appendix for a visualization of the user/item graphs, the full matrix and its singular values. It is useful in conducting controlled experiments to understand the behavior of geometry-exploiting algorithms.

---

[2]This does not say that this model is correct, just that if $M$ has the structure we assume, this iteration will capture it faithfully.

**Proposed baselines.** We report the results obtained using the following methods:

- **SGMC** and **SGMC-Z** (spectral geometric matrix completion): These are two variants of our method, differing from each other by the weights $w_{p,q}$ in (9). For these methods we chose a maximal resolution $p_{\max}, q_{\max}$ (which can be larger than $m, n$) and a skip determining the spectral resolution, denoted by $p_{\text{skip}}, q_{\text{skip}}$. SGMC uses only $w_{p_{\max}, q_{\max}} = 1$, with the rest set to zero, while SGMC-Z uses $w_{1+kp_{\text{skip}}, 1+kq_{\text{skip}}} = 1, \ k \in \mathbb{N}$.
- **DMF** (deep matrix factorization): This method, which coincides with the one suggested by Arora et al. (2019), minimizes the loss $\left\| \left( \boldsymbol{P}\boldsymbol{C}\boldsymbol{Q}^\top - \boldsymbol{M} \right) \odot \boldsymbol{S} \right\|_F^2$, i.e., it does not incorporate any geometric side information.
- **FM** (functional map): this method optimizes (10) only for $\boldsymbol{C}$.

The optimization is carried out using gradient descent with fixed step size (i.e., fixed learning rate), which is provided for each experiment alongside all the other hyper-parameters in Table 5.

**Initialization.** All our methods are deterministic and need not require multiple runs to account for initialization. We always initialize the rotation matrices $\boldsymbol{P}, \boldsymbol{Q}$ with $\alpha \boldsymbol{I}, \alpha > 0$. In Figure 7 we reported results on synthetic Netflix and ML-100K datasets for different values of $\alpha$. We noticed that for SGMC and SGMC-Z it is best to use $\alpha = 1$. According to (Arora et al., 2019), DMF requires a small $\alpha$ to decrease the generalization error (provided there are enough samples). We used DMF with $\alpha = 0.01$ for Synthetic Netflix and $\alpha = 1$ for the real world datasets, in accordance with Figure 7 and our experimentation. In the cases where only one of the bases was available, such as in Douban and Flixster-user only benchmarks, we set the basis corresponding to the absent graph to identity. The initialization of $\boldsymbol{C}$ is given by the projection of $\boldsymbol{M} \odot \boldsymbol{S}$ on the first $p_{\text{init}}$ eigenvectors of $\boldsymbol{L}_{\text{r}}$ and first $q_{\text{init}}$ eigenvectors of $\boldsymbol{L}_{\text{c}}$, i.e., $\boldsymbol{C}_0 = \boldsymbol{\Phi}_{p_{\text{init}}}^\top (\boldsymbol{M} \odot \boldsymbol{S}) \boldsymbol{\Psi}_{q_{\text{init}}}{}^3$. We use $p_{\text{init}} = q_{\text{init}} = 1$ in all SGMC experiments.

**Stopping condition.** Our stopping condition for the gradient descent iterations is based on a validation set. We use $95\%$ of the available entries for training (i.e., in the data term (9)) and the rest $5\%$ for validation. The $95/5$ split was chosen at random. We stop the iterations when the RMSE (16), evaluated on the validation set, does not change by more than $\texttt{tol} = 0.000001$ between two consecutive iterations, $|\text{RMSE}_k - \text{RMSE}_{k-1}| < \texttt{tol}$. Since we did not apply any optimization into the choice of the validation set, we also report the best RMSE achieved on the training set via early stopping. In this regard, the number of iterations is yet another hyper parameter that has to be tuned for best performance.

**Test error.** To evaluate the performance of the algorithms, we report the *root mean squared error*,

$$\text{RMSE}(\boldsymbol{X}, \boldsymbol{S}) = \sqrt{\frac{\|(\boldsymbol{X} - \boldsymbol{M}) \odot \boldsymbol{S}\|_F^2}{\sum_{i,j} \boldsymbol{S}_{i,j}}} \tag{16}$$

computed on the test set provided with each dataset. Here $\boldsymbol{X}$ is the recovered matrix and $\boldsymbol{S}$ is the binary mask representing the support of the set on which the RMSE is computed.

## 4.1 DISCUSSION

A few remarkable observations can be extracted from Table 1: First, on the Douban and ML-100K dataesets, the simple DMF shows competitive performance with all the other methods. This suggests that the geometry information is not very useful for these datasets. Second, the proposed algorithm outperforms the other methods, despite its simple and fully linear architecture. This suggests that the other methods do not exploit the geometry properly, and this fact is obscured by their cumbersome architecture. Third, while some of the experiments reported in Table 1 showed only slight margins in favor of SGMC/SGMC-Z compared to DMF, the results in the Synthetic Netflix column, the ones reported on Synthetic Movielens-100K (Table 3) and the ones reported in Figure 8, suggest that when the geometric model is accurate our methods demonstrate superior results. Table 2 presents the

---

[3] $\boldsymbol{C}_0^{p_{\text{init}} \times q_{\text{init}}}$ is the top-left submatrix of the full size $\boldsymbol{C}^{p_{\max} \times q_{\max}}$. The rest of the entries are initialized with 0.

| Model | Synthetic Neflix | Flixster | Douban | ML-100K |
|---|---|---|---|---|
| MC (Candès & Recht, 2009) | – | 1.533 | 0.845 | 0.973 |
| GMC (Kalofolias et al., 2014) | 0.3693 | – | – | 0.996 |
| GRALS (Rao et al., 2015) | 0.0114 | 1.313/1.245 | 0.833 | 0.945 |
| RGCNN (Monti et al., 2017) | 0.0053[a] | 1.179/0.926 | 0.801 | 0.929 |
| GC-MC (Berg et al., 2017) | – | **0.941**/0.917 | 0.734 | 0.910[b] |
| FM (ours) | 0.0064 | 3.32 | 3.15 | 1.10 |
| DMF (Arora et al., 2019), (ours) | 0.0468[d] | 1.06 | 0.732 | 0.918[c]/ 0.922 |
| SGMC (ours) | **0.0021** | 0.971 / 0.900 | **0.731** | 0.912 |
| SGMC-Z (ours) | 0.0036 | 0.957 / **0.888** | 0.733 | **0.907**[c]/ 0.913 |

[a] This number corresponds to the inseparable version of MGCNN.
[b] This number corresponds to GC-MC.
[c] Early stopping.
[d] Initialization with $0.01\mathbf{I}$.

Table 1: RMSE test set scores for runs on Synthetic Netflix (Monti et al., 2017), Flixster (Jamali & Ester, 2010), Douban (Ma et al., 2011), and Movielens-100K (Harper & Konstan, 2016). For Flixster, we show results for both user/item graphs (right number) and user graph only (left number). Baseline numbers are taken from (Monti et al., 2017; Berg et al., 2017).

results of Movielens-1M. First, we can deduce that a simple linear DMF model is able to match the performance of complex alternatives. Furthermore, using graphs produces slight improvements over the DMF baseline and overall provides competitive performance compared to heavily engineered methods.

**Comparing DMF and SGMC.** Arora et al. (2019) reports that for a low rank matrix, above a certain number of samples - DMF converges to the minimum nuclear norm solution (see Figure 2 in their paper). Below that number it induces a better regularization on the rank of the matrix than the nuclear norm, which still allows to recover the matrix. However, in the real datasets we tested on, the number of available samples is way below that threshold (see Table 4). The results reported in Table 1 indicate that in this extremely data poor regime the performance of DMF drops, and the extra information present in the graphs is crucial. We further confirmed this observation through a study conducted on a synthetic example, constructed from the synthetic Netflix graphs: Figure 8-left compares reconstruction errors on synthetic matrices when increasing the rank, using 15% of the samples for training. Figure 8-right compares reconstruction errors on a synthetic rank-10 matrix with increasing the density of the training set. Further details appear in the caption. The results can be summarized as follows: SGMC consistently outperforms DMF, in particular in the data poor regime. The gap between them decreases as we increase the number of samples. Similar behavior was observed in the cold-start analysis of the ML-100K dataset (see Figure 1 and the discussion below).

## 4.2 SYNTHETIC DATASETS

While the experiments reported in Table 1 showed slight margins in favor of methods using geometry, we further experimented with a synthetic model generated from the ML-100K dataset. The purpose of this experiment is to investigate whether the results are due to the DMF model or due to the geometry as incorporated by SGMC/SGMC-Z. The synthetic model was generated by projecting $M$ on the first 50 eigenvectors of $L_r, L_c$, and then matching the ratings histogram with that of the original ML-100K dataset. This nonlinear operation increased the rank of the matrix from 50 to about 400. See Figure 5 in the Appendix for a visualization of the full matrix, singular value distribution and the users/items graphs. The test set and training set were generated randomly and are the same size as those of the original dataset. The results reported in Table 3 and those on the Synthetic Netflix column in Table 1 clearly indicate that SGMC/SGMC-Z outperforms DMF, suggesting that when the geometric model is accurate it is possible to use it to improve the results. On Synthetic Netflix, we notice that by using SGMC, we outperform (Monti et al., 2017) by a significant margin, reducing the test RMSE by half. Furthermore, it can be observed that DMF performs poorly on both the synthetic datasets compared to SGMC/SGMC-Z, raising a question as to the quality of the graphs provided with those datasets on which DMF performed comparably.

| Model | ML-1M |
|---|---|
| PMF (Salakhutdinov & Mnih, 2007) | 0.883 |
| I-RBM (Salakhutdinov et al., 2007) | 0.854 |
| BiasMF (Koren et al., 2009) | 0.845 |
| NNMF (Dziugaite & Roy, 2015) | 0.843 |
| LLORMA-Local (Lee et al., 2016) | 0.833 |
| I-AUTOREC (Sedhain et al., 2015) | 0.831 |
| CF-NADE (Zheng et al., 2016) | **0.829** |
| GC-MC (Berg et al., 2017) | 0.832 |
| DMF (Arora et al., 2019), (ours) | 0.843 |
| SGMC (ours) | 0.839 |

Table 2: Comparison of test RMSE scores on Movielens-1M dataset. Baseline scores are taken from (Zheng et al., 2016; Berg et al., 2017)

| Model | Synthetic ML-100K |
|---|---|
| DMF | 0.9147 |
| SGMC | 0.5006 |
| SGMC-Z | **0.4777** |

Table 3: Comparison of average RMSE of DMF, SGMC and SGMC-Z baselines calculated on 5 randomly generated Synthetic Movielens-100K datasets.

## 4.3 COLD START ANALYSIS

A particularly interesting scenario in the context of recommender systems is the presence of *cold-start* users, referring to the users who have not rated enough movies yet. We perform an analysis of the performance of our method in the presence of such cold start users on the ML-100K dataset. In order to generate a dataset consisting of $N_c$ cold start users, we sort the users according to the number of ratings provided by each user, and retain at most $N_r$ ratings (chosen randomly) of the bottom $N_c$ users (i.e., the users who provided the least ratings). We choose the values $N_c = \{50, 100, 150, 200, 300, 400, 500\}$ and $N_r = \{1, 5, 10\}$, and run our algorithms: DMF, SGMC and SGMC-Z, with the same hyperparameter settings used for obtaining Table 1. We use the official ML-100K test set for evaluation. Similar to before, we use $5\%$ of the training samples as a validation set used for determining the stopping condition. The results presented in Figure 1 suggest that the SGMC and SGMC-Z outperform DMF significantly, indicating the importance of the geometry as data becomes scarcer. As expected, we can see that the performance drops as the number of ratings per user decreases. Furthermore, we can observe that SGMC-Z consistently outperforms SGMC by a small margin. We note that SGMC-Z, even in the presence of $N_c = 500$ cold start users with $N_r = 5$ ratings, is still able to outperform the full data performance of (Monti et al., 2017), demonstrating the strength of geometry and implicit low-rank induced by SGMC-Z.

**Ablation study.** We study the effects of different hyper-parameters of the algorithms on the final reconstruction of the matrix. We perform an ablation study on the effects of $\rho, \mu, p_{\max}, q_{\max}$ on DMF, SGMC and SGMC-Z. The results are summarized in Figures 2, 3, 4. It is interesting to note that in the case of DMF and SGMC, overparametrizing $C, Q, P$ consistently improves the performance (see Figure 4), but it only holds up to a certain point, beyond which the overparametrization does not seem to effect the reconstruction error.

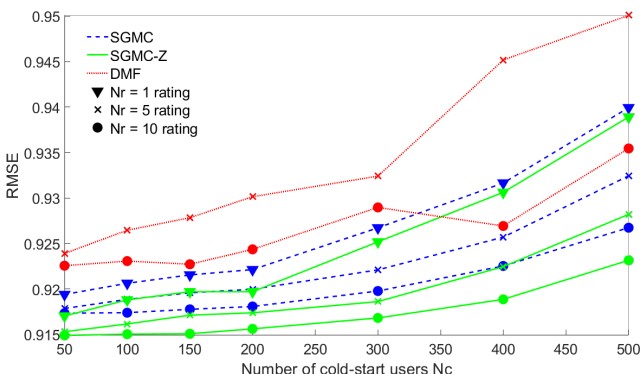

Figure 1: Comparison of test RMSE in the presence of cold start users on the ML-100K dataset. The x-axis corresponds to the number of the cold start users $N_c = 50, 100, \ldots 500$. Red, blue and green correspond to DMF, SGMC and SGMC-Z methods respectively as also shown in the legend. Different shapes of the markers indicate different number of maximum ratings ($N_r = \{1, 5, 10\}$) available per cold-start user.

**Scalability.** In this work we focused on the conceptual idea of solving matrix completion via the framework of functional maps, paying little attention to the issue of scalability. The dependence of our method on eigenvalue decomposition renders it unscalable. While this did not pose a problem for the small data sets we used in this report, it is nonetheless a drawback. We intend to address this issue in our future work.

## 5 CONCLUSION

In this work we have proposed a simple spectral technique for matrix completion, extending ideas borrowed from the field of non-rigid shape analysis. Our approach combines a full multiresolution spectral loss with (implicit) metric learning. Under a suitable change of basis, we obtain a fully linear network that gives rise to a useful interpretation via the framework of functional maps. We have demonstrated state-of-the-art results on a few recommendation systems datasets, surpassing results obtained by much more complicated architectures. In addition, we have demonstrated that some results which are usually attributed to a clever use of geometry, can be obtained without any geometry altogether. We believe that this work bridges the gap between the communities of geometric deep learning, non-rigid shape analysis, deep linear networks and spectral graph theory; it allows for a new line of research in transferring the theory and algorithms across these fields.

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

| Dataset | Users | Items | Features | Ratings | Density | Rating levels |
|---------|-------|-------|----------|---------|---------|---------------|
| Flixster | $3,000$ | $3,000$ | Users/Items | $26,173$ | $0.0029$ | $0.5, 1, \ldots, 5$ |
| Douban | $3,000$ | $3,000$ | Users | $136,891$ | $0.0152$ | $1, 2, \ldots, 5$ |
| MovieLens-100K | $943$ | $1,682$ | Users/Items | $100,000$ | $0.0630$ | $1, 2, \ldots, 5$ |
| MovieLens-1M | $6,040$ | $3,706$ | Users/Items | $1,000,209$ | $0.0447$ | $1, 2, \ldots, 5$ |
| Synthetic Netflix | $150$ | $200$ | Users/Items | $4500$ | $0.15$ | $1 \ldots 5$ [a] |
| Synthetic ML-100K | $943$ | $1,682$ | Users/Items | $100,000$ | $0.0630$ | $1, 2, \ldots, 5$ |

Table 4: Number of users, items and ratings for Flixster, Douban, Movielens-100K, Movielens-1M, Synthetic Netflix and Synthetic Movielens-100K datasets used in our experiments and their respective rating density and rating levels.

[a] The ratings are not integer-valued.

Yin Zheng, Bangsheng Tang, Wenkui Ding, and Hanning Zhou. A neural autoregressive approach to collaborative filtering. In Maria Florina Balcan and Kilian Q. Weinberger (eds.), *Proceedings of The 33rd International Conference on Machine Learning*, volume 48 of *Proceedings of Machine Learning Research*, pp. 764–773, New York, New York, USA, 20–22 Jun 2016. PMLR. URL http://proceedings.mlr.press/v48/zheng16.html.

## A  APPENDIX

### A.1  SPECTRAL SELECTION

Here we shall prove that it is enough to choose the weights $w_{p,q}$ in (9) such that each $p$ or $q$ appears only once. For simplicity we shall assume $p_{\max} = q_{\max}$ and that $p_{\text{skip}} = q_{\text{skip}} = 1$ (see Section 4).

**Proposition 1.** *Let $W$ denote the matrix of weights used in (9) and assume that they satisfy*

$$\sum_{p=1}^{p_{\max}} w_{p,q} = 1, \quad \sum_{q=1}^{q_{\max}} w_{p,q} = 1, \quad w_{p,q} \geq 0. \tag{17}$$

*Then it is always possible to choose weights $a_{p,q} \in \{0, 1\}$ achieving, for the same optimal variables, a value of (9) which is at least as low as the one obtained with $w_{p,q}$.*

*Proof.* Let $P^*, C^*, Q^*$ be the optimal variables obtained by solving (10) with some weights $W$.

Then for the same optimal variables, one can find weights $A$ achieving a value of (9) (and consequently, of (10)) which is at least as low as the one obtained with $W$, by solving

$$A^* = \arg\min_{A} \sum_{p,q} a_{p,q} Z_{p,q}(P^*, C^*, Q^*) \quad \text{s.t.} \quad \begin{cases} A\mathbf{1} & = \mathbf{1} \\ A^\top \mathbf{1} & = \mathbf{1} \\ A & \geq \mathbf{0}. \end{cases} \tag{18}$$

Problem (18) is a linear assignment problem and therefore has an integral solution $a_{p,q} \in \{0, 1\}$ (Schrijver, 1998). $\qquad \square$

| Dataset | Method | $p_{\max}/q_{\max}$ | $p_{\text{skip}}/q_{\text{skip}}$ | $\mu_r/\mu_c$ | $\rho_r/\rho_c$ | $o_r/o_c$ | Trainable variables | learning rate |
|---|---|---|---|---|---|---|---|---|
| Synthetic Netflix | DMF | 200/200 | $-/-$ | $-/-$ | $-/-$ | $-/-$ | $\boldsymbol{P,C,Q}$ | $5\times10^{-5}$ |
| | FM | 200/200 | 1/1 | 0.4/0.4 | $-/-$ | $-/-$ | $\boldsymbol{C}$ | $5\times10^{-4}$ |
| | SGMC | 20/20 | $-/-$ | 0.001/0.001 | 0.1/$-$ | $-/-$ | $\boldsymbol{P,C}$ | $5\times10^{-3}$ |
| | SGMC-Z | 500/500 | 3/1 | 0.4/0.4 | 0.1/0.1 | $-/-$ | $\boldsymbol{P,C}$ | $5\times10^{-5}$ |
| Flixster | DMF | 3000/3000 | $-/-$ | $-/-$ | $-/-$ | $-/-$ | $\boldsymbol{P,C,Q}$ | $1\times10^{-4}$ |
| | SGMC | 3000/3000 | $-/-$ | 0.0001/0.0001 | 0.0001/0.0001 | $-/-$ | $\boldsymbol{P,C,Q}$ | $1\times10^{-4}$ |
| | SGMC-Z | 200/200 | 2/2 | 0.0025/0.0025 | $-/-$ | $-/-$ | $\boldsymbol{P,C}$ | $5\times10^{-6}$ |
| Flixster (users only) | SGMC | 3000/3000 | $-/-$ | 0.0001/$-$ | 0.0001/$-$ | $-/-$ | $\boldsymbol{P,C,Q}$ | $5\times10^{-5}$ |
| | SGMC-Z | 200/200 | 20/20 | 0.0025/$-$ | 0.001/$-$ | $-/-$ | $\boldsymbol{P,C,Q}$ | $5\times10^{-7}$ |
| Douban | DMF | 3000/3000 | $-/-$ | $-/-$ | $-/-$ | $-/-$ | $\boldsymbol{P,C,Q}$ | $6\times10^{-6}$ |
| | SGMC | 2500/2500 | $-/-$ | 0.001/$-$ | 0.001/$-$ | 0.001/$-$ | $\boldsymbol{P,C,Q}$ | $2\times10^{-6}$ |
| | SGMC-Z | 1000/1000 | 50/1000 | 0.011/0 | 0.004/0 | 0.001/0 | $\boldsymbol{P,C,Q}$ | $2\times10^{-6}$ |
| ML-100K | DMF | 2000/2000 | $-/-$ | $-/-$ | $-/-$ | $-/-$ | $\boldsymbol{P,C,Q}$ | $5\times10^{-5}$ |
| | SGMC | 4000/4000 | $-/-$ | 0.0003/0.0003 | 0.0001/0.0001 | $-/-$ | $\boldsymbol{P,C,Q}$ | $5\times10^{-5}$ |
| | SGMC-Z | 3200/3200 | 30/35 | 0.03/0.03 | 0.2/0.2 | 0.09/0.09 | $\boldsymbol{P,C,Q}$ | $3\times10^{-7}$ |
| ML-1M | DMF | 7000/7000 | $-/-$ | $-/-$ | $-/-$ | $-/-$ | $\boldsymbol{P,C,Q}$ | $1\times10^{-5}$ |
| | SGMC | 7000/7000 | $-/-$ | 0.0001/0.0001 | $-/-$ | $-/-$ | $\boldsymbol{P,C}$ | $8\times10^{-5}$ |
| Synthetic ML-100K | DMF | 8000/8000 | $-/-$ | $-/-$ | $-/-$ | $-/-$ | $\boldsymbol{P,C,Q}$ | $9\times10^{-5}$ |
| | SGMC | 600/600 | $-/-$ | 0.001/0.001 | 0.009/0.009 | $-/-$ | $\boldsymbol{P,C,Q}$ | $2\times10^{-5}$ |
| | SGMC-Z | 500/500 | 3/1 | 0.001/0.001 | 0.009/0.009 | $-/-$ | $\boldsymbol{P,C}$ | $5\times10^{-6}$ |

Table 5: Hyper-parameter settings for the algorithms: DMF, SGMC and SGMC-Z, reported in Tables 1, 2, 3.

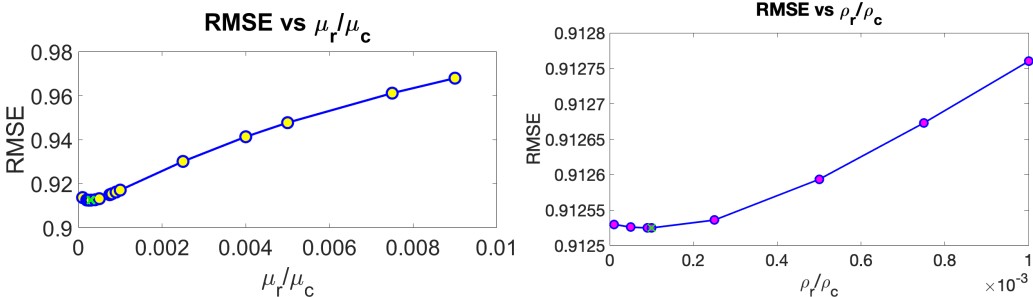

Figure 2: Ablating $\rho_r = \rho_c$ and $\mu_r = \mu_c$ of SGMC on the ML-100K dataset. The rest of the parameters were set to the ones reported in Table 5. Green X denotes the baseline from Table 1.

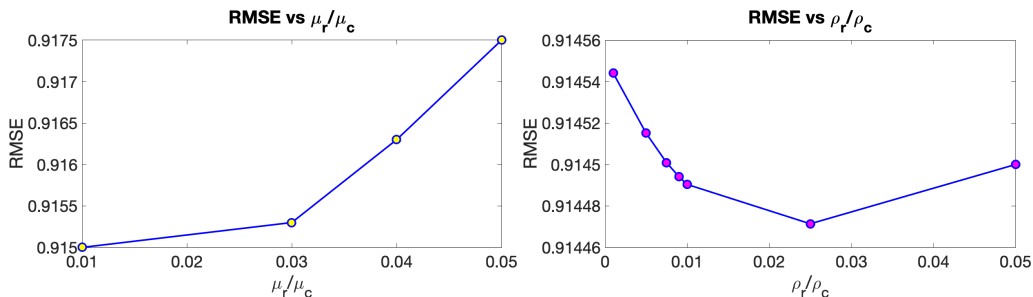

Figure 3: Ablating $\rho_r, \rho_c$ and $\mu_r, \mu_c$ of SGMC-Z on the ML-100K dataset. The rest of the parameters were set to the ones reported in Table 5.

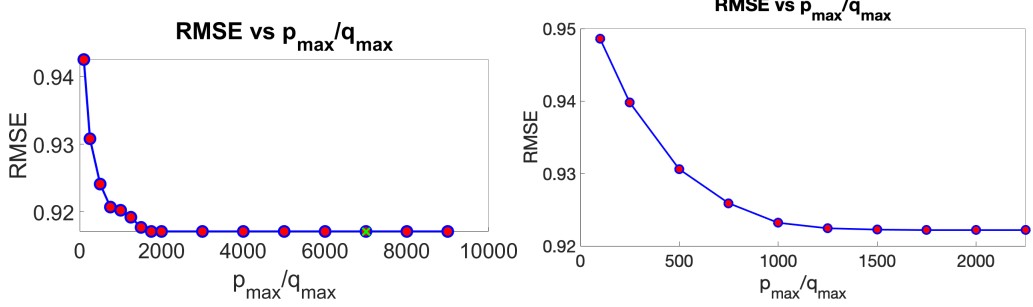

Figure 4: Effect of overparametrization: SGMC (left) and DMF (right). x-axis indicates the values of $p_{\max}, q_{\max}$, and y-axis presents the RMSE. Green X denotes the baseline from Table 1.

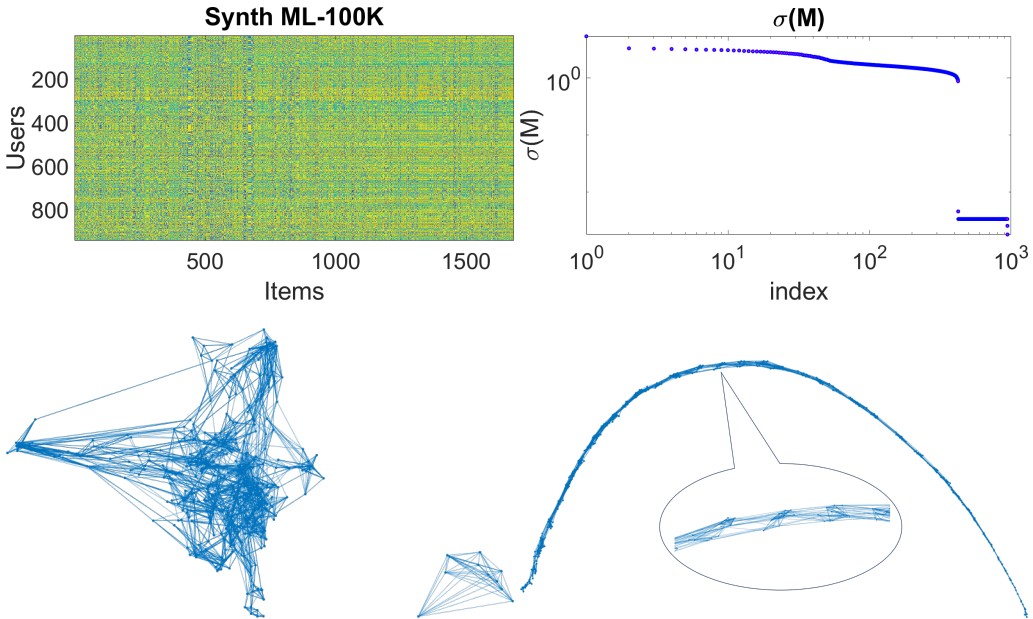

Figure 5: Synthetic Movielens-100k. Top-left: Full matrix. Top-right: singular values of the full matrix. Bottom left & right: items & users graph. Both graphs are constructed using 10 nearest neighbors.

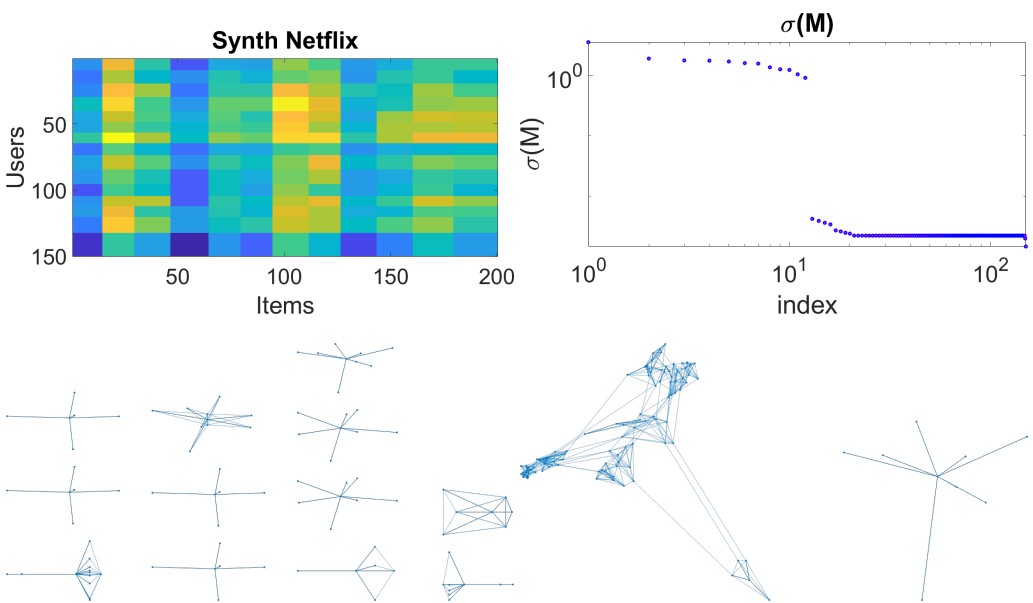

Figure 6: Synthetic Netflix. Top-left: Full matrix. Top-right: singular values of the full matrix. Bottom left & right: items & users graph. Taken from (Monti et al., 2017).

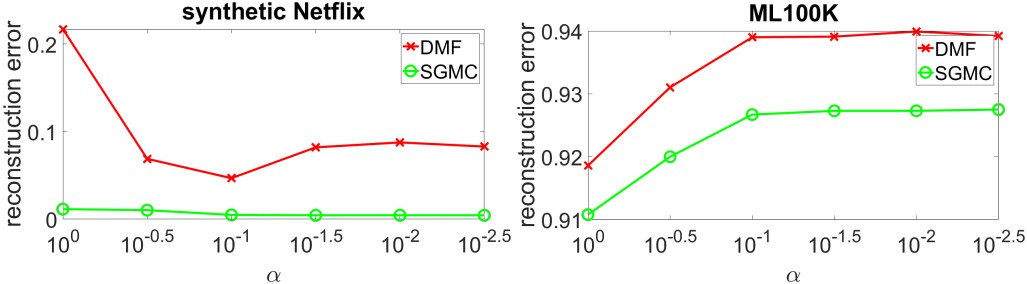

Figure 7: Reconstruction error (on the test set) vs. scale of initialization. For each method we initialized $P, Q$ with $\alpha I$. SGMC consistently outperforms DMF for any initialization.

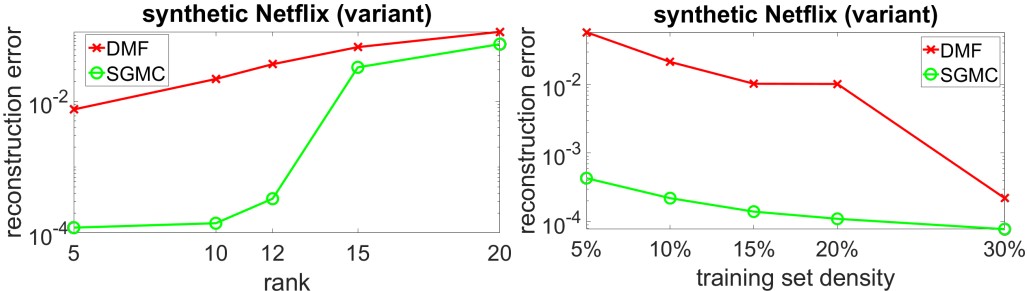

Figure 8: In these experiments we generated matrices using the synthetic Netflix graphs to test the dependence of SGMC and DMF on the rank and on the number of training samples. Left: Reconstruction error (on the test set) vs. rank. As the rank increases, the reconstruction error increases, but it increases slower for SGMC than for DMF. For the training set we used $15\%$ of the points chosen at random (same training set for all experiments). Right: Reconstruction error (on the test set) vs. size of the sampling set, for a random rank 10 matrix. As we increase the number of samples, the gap between DMF and SGMC reduces. Still, even when using $30\%$ of the samples, SGMC performs better for the same number of iterations. In all experiments we used the following hyperparameters: $\mu_r = \mu_c = 0.001, \rho_r = \rho_c = 0.001, \texttt{maxiter} = 8 \times 10^6$ .

