# OpenReview forum: "Deep geometric matrix completion:  Are we doing it right?"
_ICLR.cc/2020/Conference — Reject_

### Official Review · AnonReviewer2 · 2019-10-21
**Official Blind Review #2**

**Rating:** 6

**Review:**

This paper proposes a novel approach for the loss function of matrix completion when geometric information is available. The proposed method consists of two ideas: (1) spectral regularization (i.e., Dirichlet energy) with a re-parameterizing basis and (2) multiresolution of spectral loss (i.e., zoomout loss). In addition, the zoomout loss is motivated by the approach for shape correspondence and can be a generalization of the recent matrix completion method (deep matrix factorization). Empirical results show the best performance compared to other recent methods under small-scale datasets. Moreover, the proposed method outperforms when the geometric model is accurate (verified on the synthetic setting) and this can reflect that the proposed method is a good choice when the graph structures are given.

This work can be a significant contribution as it is a simple linear model but practically performs better than other deep nonlinear networks (e.g., RGCNN). Additionally, the proposed loss functions utilize only the spectral information of graph structure with novel approaches. However, there are some drawbacks to this work. First, it requires a good quality of geometric model which is hard to obtain in practical datasets. Second, the proposed method has a scalability issue since it requires eigendecompositions of graph Laplacians (as discussed in the paper). This can be a problem for real and large-scale datasets.

Overall, this paper presents a novel approach utilizing graph spectral information with empirical improvements. But, I vote for weak acceptance due to its drawbacks as mentioned above.

Main concerns:

1. It is not clear why minimizing Dirichlet energy can improve the performance of matrix completion. In the paper, the authors mention that it promotes smooth functions on the graph nodes, but not fully clear why smooth functions are good. And how much does the accuracy increase (or decrease) when using the Dirichlet regularization?

2. Authors argue that the re-parameterizing of the basis (emerging P and Q) can find a better geometric model (section 2). So, it is expected that the proposed method shows a better result when the given geometric model is not accurate. However, the empirical results are reported poor improvements for inaccurate geometric models. Does this make sense?

For experiments:

1. What is the number of trainable parameters for each method? Since the proposed method is overparameterized, it is not clear that the empirical improvements come from the overparameterizing or the proposed loss function. It would be great to report the number of parameters of all other methods by setting similar numbers.

2. It is not clear how to generate the synthetic dataset, i.e., projecting a random matrix on te the first few eigenvectors of L_r and L_c. It would be better to give more details.

3. What are the training times of the proposed method and other competitors?

4. Why results of FM are not reported under other datasets?

Minor comments:

1. In page 4, please edit “We explore The” -> “We explore the”.

2. In equation (15), writing “\odot S” twice seems to be unnecessary.


**Experience Assessment:**

I have published one or two papers in this area.

**Review Assessment: Checking Correctness Of Derivations And Theory:**

I carefully checked the derivations and theory.

**Review Assessment: Checking Correctness Of Experiments:**

I carefully checked the experiments.

**Review Assessment: Thoroughness In Paper Reading:**

I read the paper thoroughly.

---

> ### Author Response · Authors · 2019-11-12
> **Reply to reviewer #2 (2/2)**
>
> Experimentation:
>
> (1)  The number of trainable parameters for our method is the number of elements in $\mathbf{P},\mathbf{C},\mathbf{Q}$. This is chosen according to $p_{\mathrm{max}},q_{\mathrm{max}}$ - a hyperparameter in our setting reported in Table 5. Overparameterization alone is not enough to produce the empirical improvements we reported. Without explicit regularization it would overfit the training data and perform poorly on the test data, specifically in the data poor regime. Also see comment (3) of reviewer #3 and the answer we provided.
>
> Regarding the other methods: we omitted those details as it is not clear how to compare the number of parameters between methods of ostensibly different nature, and it wasn’t the focus of our paper.
>
> (2) To generate the Synthetic ML-100K dataset we did the following:
>
>  - We computed the first $k=50$ eigenvectors $\mathbf{\Phi}_k,\mathbf{\Psi}_k$ of $\mathbf{L}_\mathrm{r}$ and $\mathbf{L}_\mathrm{c}$, the Laplacians of the row and column graphs for the ML-100K dataset
>
> - We projected $\mathbf{M}$ on this subspace, i.e.,
> \[
> \mathbf{M}_{\mathrm{proj}} =\mathbf{\Phi}_k\mathbf{\Phi}_k^\top\mathbf{M}\mathbf{\Psi}_k\mathbf{\Psi}_k^\top;
> \]
> - We performed histogram matching between $\mathbf{M}_\mathrm{proj}$ and $\mathbf{M}$ such that the histogram of entries in $\mathbf{M}_\mathrm{proj}$ is the same as in $\mathbf{M}$. This nonlinear operation increases the rank of the matrix, so it is no longer k, but the perturbation to the singular values is small (verified empirically).
>
> (3) We do not have the training times for the other methods as we took the results from the corresponding papers. Our method (SGMC) runs in just a few minutes, depending on the datasets. For example, on ML-100K with the parameters reported in table 5 it takes about a minute, including the eigendecomposition and excluding the time taken to build the computational graph in Tensorflow.
>
> (4) The results of the FM method are poor for the other datasets so we did not include them. We will include them in the revised version.
>
> Minor comments:
>
> (2) Regarding equation (15):  $\odot S$ should appear twice, following from the computation of the gradient. If $S$ is binary (as in our case) then $S\odot S = S$ and one of the $S$ disappears.
>
> We again thank you for the suggestions and will incorporate your comments into the revised version of the paper.
>
> Yours sincerely,
> The authors.

---

> ### Author Response · Authors · 2019-11-12
> **Reply to reviewer #2 (1/2)**
>
> Dear reviewer #2,
>
> Thank you for your comments! In what follows, we will try to address in detail the issues you raised:
>
> (0) Regarding scalability of the spectral decomposition: This is an issue we acknowledged in the paper. Despite this shortcoming, our method is useful for a variety of mid-size problems, and we believe that the scalability issue does not take away from its theoretical and practical merits.
>
> Regarding the quality of the available geometric model:
>
> First, one of the main purposes of our paper is to reflect on the quality of geometric matrix completion (GMC) methods. We raise the question whether the models used by GMC methods are truly useful, or their results are due to some underlying phenomena (e.g., an implicit regularization inherent in gradient descent)? Indeed, our experimentation shows that results on par with state-of-the-art GMC can be obtained even with simple baselines such as deep matrix factorization (Arora et. al). We believe that these observations following from our extensive experimentation provide an important contribution to the community.
>
> Second, it is not always difficult to obtain accurate geometric models. As we acknowledge in the paper, our intuition and inspiration comes from shape analysis. In this field the geometric models are usually quite accurate. Focusing on such cases allowed us to develop the geometric intuition behind our method and attribute its success (or failure) to the quality of the available geometric model.
>
> Third, there is some merit in starting with a crude geometric model and refining it with our optimization. This crude geometric model can be obtained via pre-processing, e.g., a supervised learning approach, and refined on the actual data. In the following link we provide a toy example that allows to investigate what happens when there is a mismatch between the graphs used to generate the data, and the graphs used to estimate it. There is a clear evidence that learning the graphs (i.e., by “rotating” the Laplacian bases) improves the estimation, (compared to using the graphs as is)
>
> https://colab.research.google.com/drive/1OkNEiTHok14gcVf3NxFIbAFutDN6-Tx6
>
> (1) Our model tacitly assumes a particular form of an (approximate) low rank matrix:
> A matrix that is composed of a linear combination of the first harmonic vectors of some product graph (i.e., those corresponding to low frequencies). We further simplified this assumption and assumed this matrix is smoothed separately on the rows graph and the columns graph.
> Why is this a good model? We believe that information about a particular user can be shared across other similar users (as captured by the graph edges) via a process of smoothing (or diffusion). Following this process, similar users should give similar ratings. In the same way, similar items should have similar ratings. This similarity across neighbouring nodes is captured by the Dirichlet energies for the rows and column graphs - small Dirichlet energy corresponds to a smooth function. Please also refer to the explanation we provided to reviewer #1.
> Using this model is particularly helpful (empirically) in the data poor regime which most real data sets lie in. In this regime it is not enough to assume just that the matrix is low rank, despite methods like DMF (Arora et. al) exploiting complex rank regularizations. An additional assumption on the structure of this matrix turns out to be very helpful, provided that a geometric model (even a mildly crude one) is available. See our cold start analysis (FIgure 1) that explores this regime and our reply to reviewer #3. You are also welcome to play with the number of samples in the aforementioned link and witness the effectiveness of our method in the data poor regime.
>
> (2) The fact that our model only provides a marginal improvement in the case of a poor geometric model definitely makes sense to us. When the geometric model is poor, there is a small difference (if at all) between using the graphs or not using them altogether. In these cases there is no clear evidence as to what is the contributing factor to the rank regularization - be it an implicit regularization due to the gradient descent or some other factor. For the Douban dataset for example (table 1) we see that the DMF method (Arora et. al) is competitive with the other more complicated methods. In our opinion, this is due to poor geometry. We further explored this phenomena (i.e., by perturbing the graphs with increasing noise) in the toy example in the aforementioned link.

---

### Official Review · AnonReviewer1 · 2019-10-24
**Official Blind Review #1**

**Rating:** 3

**Review:**

This paper proposes a new method for geometric matrix completion based on functional maps. The proposed algorithm is a simple shallow and fully linear network. Experimental results demonstrate the effectiveness of the proposed method.

The proposed method is new and has been shown good empirical results. The paper also points out a new way to interpret matrix completion. On the other hand, the proposed method seems ad hoc and there is no clear evidence why it is better than other baselines except the empirical results. The paper also has some clearance issues, making it hard to understand. I vote for a weak reject of the paper at the current pace and would like to increase my score if the following questions can be clearly answered.

1.	Why do we need to propose the algorithm? Is it because we have the functional maps technique motivated from shape correspondence, and we can see some connection of such technique with matric completion? If it is true, we surely can have a new algorithm based on such a new technique. But I can still not understand why the method work, at least, in an intuitive way.
2.	What is the sample complexity of the proposed matrix completion algorithm?
The introduction of the paper is poorly written. The first paragraph and the third one both contain some introduction to matric completion, which results in a lot of redundant information. The second paragraph and the fourth one are redundant in the same way since they both focus on geometric matrix completion. I think besides introducing what is matrix completion and what is geometric completion, the introduction part should focus more on the motivation to propose the algorithm. However, I can only see from the end of the second paragraph (some simple models need to be proposed) and the fifth paragraph (“The inspiration of our paper”) some motivation information. The introduction part needs to be re-organized to provide more useful information about the paper rather than a literature review.

There is some unclear/inaccurate/subjective statement in the introduction part. For example, “Self-supervised learning” needs a reference. Why geometric matrix completion generalizes the standard deep learning approaches is not clear. What does it mean by “their design is … cumbersome and non-intuitive”? The shape correspondence is never explained until very later in the paper.  Also, there are some unclear issues besides the Introduction part. For example, what does it mean by “the product graph”? All these issues need to be clarified before the paper can be accepted.

---------------------------------------------------
Thank you for the detailed rebuttal. For Q1, it clearly explains how does the method work. However, it is still not clear why does the method work. I also have another concern after reading the rebuttal, if the shape correspondence is not that important, why make it an important motivation in the paper? For Q2, it is interesting to see some theoretical results on the sample complexity, rather than an experimental one. The paper would also be much better if the clearance issues can be addressed. Even if I would not vote for an accept this time, I am looking forward to a revised version in the future.


**Experience Assessment:**

I have published one or two papers in this area.

**Review Assessment: Checking Correctness Of Derivations And Theory:**

I assessed the sensibility of the derivations and theory.

**Review Assessment: Checking Correctness Of Experiments:**

I assessed the sensibility of the experiments.

**Review Assessment: Thoroughness In Paper Reading:**

I read the paper at least twice and used my best judgement in assessing the paper.

---

> ### Author Response · Authors · 2019-11-12
> **Reply to Reviewer #1 (2/2)**
>
> Another interesting observation we made is that our method is essentially an overparameterized deep matrix factorization (DMF) method with some additional structure. DMF has been proven recently (see Arora et al. 2019 and the discussion with reviewer #3) to promote a low rank via implicit regularization of gradient descent. This is a contributing factor to the success of our method.
>
> We made up a tutorial to allow experimenting with our method in the link below, and we hope you can find it useful to understand the method better:
>
> 	https://colab.research.google.com/drive/1OkNEiTHok14gcVf3NxFIbAFutDN6-Tx6
>
> (2) The number of available ratings for each dataset is provided in Table 4. If by “sample complexity” you mean how the test error changes with the size of the training set, we believe that our cold start analysis (Figure 1) provides an answer: we show that the SGMC-Z version of our method is particularly more effective in the data-poor regime than the SGMC and other competing algorithms. Even after retaining only 5 ratings for more than half the users, we still get competitive results compared, for example, to RGCNN (compare Figure 1 and Table 1).
>
> Regarding the presentation issues: we thank you for the suggestions and will reformulate some parts of the paper according to your recommendations.
>
> Yours sincerely,
> The authors.

---

> ### Author Response · Authors · 2019-11-12
> **Reply to reviewer #1 (1/2)**
>
> Dear reviewer #1,
>
> Thank you for your comments! In what follows, we will try to address in detail the issues you raised:
>
> Our method comes from geometric considerations rather than an ad-hoc construction. We advocate that focusing on the geometric interpretation can sometimes lead to simplified architectures, therefore the title of our paper. In our case, it results in a fully linear network which, in our humble opinion, is a simpler architecture compared to some other competing geometric matrix completion methods. This motivates our subjective claims regarding “cumbersome and non-intuitive designs”. The message we were trying to convey was that architectural designs that originated in Euclidean deep learning such as convolutional layers followed by pointwise non-linearities, might not be the best candidates for other domains. For example, Wu et al. 2019 showed that it is possible to simplify  graph neural network architecture with a minor compromise to the end task. We will try to clarify these claims and address your major concerns below:
>
> (1) Our inspiration for the method came from problems in shape correspondence. A correspondence problem is a matrix completion problem with some constraints on the matrix. Although it served as an inspiration, it is unnecessary to understand the correspondence problem on shapes in order to understand our method. An intuitive explanation can be given along the following lines:
>
> We are given a rating matrix, where each entry in the matrix is the rating given by a user i to an item j. Our model assumes that similar users should rate items similarly, and similar items should be rated similarly by different users. Similarity between users/items is encoded by some external graphs (constructed, respectively, on the row/column spaces of the matrix).
> On a graph, one can define a function, i.e., a vector $\mathbf{x}$ whose entries are values on the nodes of the graph. With some abuse of proper mathematical terminology, we call the function “smooth” if its values on adjacent nodes are close. This kind of smooth behaviour is encoded by the projection of the function on the first eigenvectors of the graph Laplacian $\mathbf{L}$, in the same way that a “smooth” function in Euclidean space is composed only of harmonic functions with small frequencies. The eigenbasis of a graph Laplacian is the graph analogue of the Euclidean Fourier basis.
> Since we have two graphs, we have two such Fourier bases, $\mathbf{\Phi},\mathbf{\Psi}$, and we can treat the rating matrix as an outer product of two functions: one defined on the users graph and one defined on the items graph. Each one of these functions is smooth in its own right.  We therefore write our matrix as $\mathbf{X} = \mathbf{\Phi}\mathbf{C}\mathbf{\Psi}^\top$.
>
> The Dirichlet energy,
> \begin{equation}
>     \mathbf{x}^\top \mathbf{L}\mathbf{x} = \sum_{(a,b)\in E}\omega_{a,b}\left(x(a)-x(b)\right)^2,
> \end{equation}
> penalizes the difference between the function values on adjacent nodes, and therefore minimizing it promotes such smooth functions. So, in principle, one can find smooth functions on a graph by minimizing some data term (e.g., the L2 norm) and regularize it with some smoothness term such as the quadratic Dirichlet energy. This gives rise to a simple convex problem which can give great results if the graphs are accurate,
>
> (equation 1)
> \begin{equation}
> \min_{\mathbf{C}} \|\left(\mathbf{\Phi}\mathbf{C}\mathbf{\Psi}^\top-\mathbf{M}\right)\odot \mathbf{S}\|_F^2 + \mu_rE_{Dirichlet}^r(\mathbf{C})+\mu_cE_{Dirichlet}^c(\mathbf{C}).
> \end{equation}
> Unfortunately, our graphs are inaccurate since it is hard to model the relationship between users and items. Nevertheless, we would like to enforce our matrix to follow the model described above due to its simplicity, despite the inaccuracies in the graphs. To do that, we assume that the graphs can be “corrected”, and on the “corrected” graphs the matrix will still be smooth. Since correcting the graphs seems like a hard task, and anyway we are only interested in the representation of the function in the eigenbases of the “corrected” graph Laplacians, we try to directly get these bases $\mathbf{\Phi}_{new},\mathbf{\Psi}_{new}$ by applying a linear transformation to the old bases: $\mathbf{\Phi}\mathbf{P},\mathbf{\Psi}\mathbf{Q}$.
> We only need to make sure that the new bases are indeed Laplacian eigenbases, and this can be done by requiring them to diagonalize the new Laplacians. Since we don’t have the new Laplacians, we will use the old ones as proxies.
>
> If all is working according to plan, we will end up with new graphs (which remain latent), on which our unknown matrix  $\mathbf{X}=\mathbf{\Phi}\mathbf{P}\mathbf{C}\mathbf{Q}^\top\mathbf{\Psi}^\top$
> is smooth, and therefore has a low Dirichlet energy. This entire story is captured by equation (10) in our paper, with some additional minor details.

---

### Official Review · AnonReviewer3 · 2019-10-26
**Official Blind Review #3**

**Rating:** 3

**Review:**

This paper aims to solve the matrix completion problem by incorporating geometric information. The proposed approach involves using graphs encoding relations between rows (and columns), applying spectral decomposition to these graphs, and using a multi-resolution spectral geometric loss to reconstruct the functional map which could then be used to directly recover the underlying matrix. The paper evaluates the proposed network on both synthetic and real datasets and shows improvements over the existing geometric methods and convex relaxations.

While the geometric approach looks interesting and the experimental results seem promising, it is unclear why the proposed approach works, and the comparison with [Arora et al. (2019)] is not fair. Below are the specific comments.

(1) The proposed approach (formulation (10)) involves too many parameters (including the weights w in (9)) that need to be tuned. The authors should discuss how to select the parameters after (10). This also raises the question of how practical the proposed approach is.

(2) The authors claim the first contribution is to provide the geometric interpretation of deep matrix factorization via the functional maps framework. However, I didn't see clearly the interpretation. If it refers to the parametrization of X by \Phi P C Q^T \Psi^T, then it is just a special case of deep matrix factorization since both \Phi and \Psi are fixed, and P and Q are optimized to be approximately orthonormal.

(3) Due to over-parameterization, in general deep matrix factorization would suffer from overfitting. That being said [Gunasekar et al. (2017), Arora et al. (2019)] prove that gradient descent induces implicit regularization if the algorithm is initialized with factors that are very "small". However, in the experiments, both P and Q are initialized as the identity, which is not close to zero. Indeed, it was proved in the following paper that the generalization gap will be proportional to the energy of the initialization, even for matrix factorization.

Yuanzhi Li, Tengyu Ma, and Hongyang Zhang, Algorithmic Regularization in Over-parameterized Matrix Sensing and Neural Networks with Quadratic Activations.

(4) As a followup question, without such implicit regularization, it is unclear why the proposed approach does not suffer from overfitting. A discussion along this line is required. Though the authors include the connection between [Arora et al. (2019)], this is not convincing enough since as explained above, the implicit regularization there depends on the smallness of the initialization.

**Experience Assessment:**

I have published in this field for several years.

**Review Assessment: Checking Correctness Of Derivations And Theory:**

I carefully checked the derivations and theory.

**Review Assessment: Checking Correctness Of Experiments:**

I assessed the sensibility of the experiments.

**Review Assessment: Thoroughness In Paper Reading:**

I read the paper thoroughly.

---

> ### Author Response · Authors · 2019-11-12
> **Reply to Reviewer #3**
>
> Dear reviewer #3,
>
> Thank you for your comments! In what follows, we will try to address in detail the issues you raised:
>
> (1) We believe this is a misunderstanding of the hyperparams involved. While we stated in the paper the full scope of possible hyperparams in this general framework, we limited ourselves to only two settings:
>
> (a) SGMC - In this setting we set the weights $w_{ij}$ to 0 at all resolutions except the last one (full resolution).
>
> (b) SGMC-Z - In this setting we chose a-priori some spectral skip parameters (p_skip, q_skip) and we set $w_{ij}=1$ for (i=1+k*p_skip, j=1+k*q_skip), i.e., we sample the parameter space (p,q) on a grid with spacing (p_skip, q_skip), and set $w_{ij}=1$ only on the diagonal of this grid.  We did not try to explore any other setting for $w_{ij}$.
>
> Overall, the number of hyperparams involved is between 8 to 10 (2 for each energy involved and the p_max/q_max, p_skip/q_skip params). Moreover, we usually define the same hyperparams for the rows and columns energies, so it is about half that number. In a future work we will also make some of these parameters such as p_skip/q_skip learnable.
>
> Also note that, following our ablation study, the dependence on the hyperparams is quite small (on some even negligible), and it is rather easy to tune them using a validation set.
>
> (2) As we noted, the data term of the SGMC is a special form of DMF from Arora et. al. But we also introduced two important additional terms:
>
> A Dirichlet energy term - promoting smoothness on the (inferred) graphs.
>
> A diagonalization term - promoting the new (inferred) bases to be Laplacian eigenbases.
>
> These three terms together provide the geometric interpretation: If we treat the two factors $P,Q$ as corrections to some harmonic bases, and approximately enforce those new bases to also be harmonic bases (i.e., approximately diagonalizing the corresponding graph Laplacians), then we can model our matrix as some approximately bandlimited signal on a new product graph, whose functional space can be spanned by the new bases.
>
> Following your remark, we acknowledge there is some lack of clarity in the way we presented our approach: We do not provide a geometric interpretation to DMF but rather embed it within a bigger geometric framework. Once the aforementioned two terms are included, the geometric interpretation emerges.
>
> (3) We thank you for raising this point. Our intention in including a comparison to DMF was to show how a simple method such as DMF can produce results on par with state-of-the-art geometric methods. This is one of the main messages of our paper - to show how badly the underlying geometry is being used (or how bad is the geometry being used) in geometric matrix completion methods.
> Following your remark, we performed the experiments with DMF again, using a “small” initialization, and indeed we got a large improvement on the synthetic datasets! On the real datasets, however, we did not observe any improvement. The experiments we report in the following link measure the reconstruction error achieved by each method when the initialization is scaled by $10^{-\alpha}$, $\alpha>0$, as suggested in Li et al.
>
> https://drive.google.com/open?id=1pduAXS_NHwC1DhornDD9A78YehwCAjzR
>
> (4) Our regularization is explicit. It follows from both the Dirichlet energy and the multiresolution loss which weighs more heavily the lower frequency part of the functional map, as explained in the text. In order to better illustrate why the approach works, we came up with a toy example that can be found in the following link:
>
> https://colab.research.google.com/drive/1OkNEiTHok14gcVf3NxFIbAFutDN6-Tx6
>
> This regularization does not necessitate depth, as in DMF, but still allows to enjoy the implicit regularization inherent to DMF with gradient descent methods.
>
> We have a compelling explanation for the better performance of our method compared to DMF: As Arora et al reports (see Figure 2 in their paper), above a certain number of samples, DMF converges to the minimum norm solution. Below that number it induces a better regularization on the rank of the matrix, which still allows to recover low rank matrices. However, in the real datasets we tested on, the number of available samples is way below that threshold, as the rank of those matrices is not that low (See Table 4 in our paper). In this extremely data poor regime, DMF performs poorly, and the extra information present in the graphs is crucial. This is consistent with our experimentation with the toy problem we shared in the link above, and you can test yourself by changing the number of training samples. For a rank-10 matrix, using more than 30% of the entries allows for a very low reconstruction error with DMF, which outperforms our method (by a small margin). However, when going below 20%, our method demonstrates a clear advantage. We will add a discussion along these lines with relevant plots to the revised version.
>
> Yours sincerely,
> The authors.

---

### Public Comment · ~Abhishek_Sharma1 · 2019-10-02
**Ablation study and source code**

Dear Authors,

This is a very interesting work. Could you please comment on:

a) missing ablation study of various terms in Eq. 10. In particular, how critical are the Dirichlet energy terms in the overall performance.

b) difference in diagonalization proposed here and in Coupled quasi harmonic basis (Kovnatsky et al.)


c) Do you have plans to release the source code.

---

> ### Author Response · Authors · 2019-10-02
> **re: Ablation study and source code**
>
> Thank you for your interest in our paper!
>
> a). We ablated all but the coefficients for the orthogonalization terms.  Figure 2 shows the ablation study for Dirichlet (left) and Diagonalizaition (right) terms for SGMC,
> and Figure 3 shows the ablation study for Dirichlet (left) and Diagonalizaition (right) terms for SGMC-Z.
>
> We can summarize the importance of the different terms as follows:
>
> Dirichlet - seems to be rather important.
> Diagonalization - seems to be moderately important.
> Orthogonalization - hardly important (that's why we did not include it in our ablation study).
>
> It should be emphasized that the Dirichlet energy is with respect to the new basis and not the old basis (please refer to equation 7).
> You can also look at the hyperparameters table (table 5) and and see that the Dirichlet energy was needed.
>
> b). The diagonalization energy proposed in Coupled quasi harmonic basis (Kovnatsky et al.) is essentially the same as ours. The differences are due to notation:
> We define off(A) as the off-diagonal entries of A, whereas Kovnatsky et al. define off(A) as the sum of squares of the off-diagonal entries of A.
> In our notation, that would be ||off(A)||^2 = sum(i~=j) a^2_{ij}.
> Also,  Kovnatsky et al. enforces orthogonality (w.r.t to the manifold inner product), whereas we only promote approximate orthogonality via a penalty function.
> It should be noted that we are not the first to propose functional map estimation via joint diagonlization. These ideas have been spinning around for some time in the shape analysis community, and we borrowed them for the problem of matrix completion. We acknowledge Litany et. al's  "Fully Spectral Functional Maps", the most up to date shape correspondence method based on joint diagonalization, as our source of inspiration.
>
> c). We shall provide a link to the source code soon. Stay tuned!
>
> The authors

---

> > ### Public Comment · ~Abhishek_Sharma1 · 2019-10-03
> > **Re: Source code**
> >
> > Thank you for the quick reply. Looking forward to the code!

---

### Author Response · Authors · 2019-11-15
**Revised paper**

Dear reviewers & ACs,

We have uploaded a revised version of our paper according to the discussions below.

The following updates have been made:

(1) We improved the introduction, eliminated redundancies and added some motivation.
(2) We updated the paper’s main contributions.
(3) We added the missing definitions, i.e., product graph.
(4) We included a study on the effect of the scale of the initialization.
(5) We filled in the missing results for the FM method and updated the results obtained with DMF to account for the initialization.
(6) We included a study on the effect of the rank and the number of samples on the reconstruction error for both DMF and SGMC.
(7) We added a discussion summarizing the results of these studies. In particular, conjecturing that the superb performance of SGMC is related to the fact that we work in an extremely data-poor regime and therefore the implicit regularization of DMF is not enough.
(8) We included a link to a simplified version of the code in the form of a jupyter notebook.

We thank you for your time reviewing our paper and for the useful comments. It helped us improve our paper and reconfirm our reportings.

Regards,
The authors.

---

### Decision · Program_Chairs · 2019-12-19

**Decision:**

Reject

**Comment:**

This paper proposes a multiresolution spectral geometric loss called the zoomout loss to help with matrix completion, and show state-of-the-art results on several recommendation benchmarks, although experiments also show that the result improvements are not always dependent upon the geometric loss itself.
Reviewers find the idea interesting and the results promising but also have important concerns about the experiments not establishing how the approach truly works. Authors have clarified their explanations in the revisions and provided requested experiments (e.g., on the importance of the initialization size), however important reservations re. why the approach works are still not sufficiently addressed, and would require more iterations to fulfill the potential of this paper.
Therefore, we recommend rejection.